**Data Availability Statement:** All relevant data are within the paper and its Supporting information files.

**Funding:** The author(s) received no specific funding for this work.

# Forensic analysis of the Turkey 2023 presidential election reveals extreme vote swings in remote areas

**Peter Klimek**[1,2,3]*, **Ahmet Aykaç**[4], **Stefan Thurner**[1,2,3,5]

**1** Section for Science of Complex Systems, CeDAS, Medical University of Vienna, Vienna, Austria, **2** Complexity Science Hub Vienna, Vienna, Austria, **3** Supply Chain Intelligence Institute Austria (ASCII), Vienna, Austria, **4** Theseus International Management Institute, Sophia Antipolis, France, **5** Santa Fe Institute, Santa Fe, NM, United States of America

* peter.klimek@meduniwien.ac.at

## Abstract

Concerns about the integrity of Turkey's elections have increased with the recent transition from a parliamentary democracy to an executive presidency under Recep Tayyip Erdoğan. Election forensics tools are used to identify statistical traces of certain types of electoral fraud, providing important information about the integrity and validity of democratic elections. Such analyses of the 2017 and 2018 Turkish elections revealed that malpractices such as ballot stuffing or voter manipulation may indeed have played a significant role in determining the election results. Here, we apply election forensic statistical tests for ballot stuffing and voter manipulation to the results of the 2023 presidential election in Turkey. We find that both rounds of the 2023 presidential election exhibit similar statistical irregularities to those observed in the 2018 presidential election, however the magnitude of these distortions has decreased. We estimate that 2.4% (SD 1.9%) and 1.9% (SD 1.7%) of electoral units may have been affected by ballot stuffing practices in favour of Erdoğan in the first and second rounds, respectively, compared to 8.5% (SD 3.9%) in 2018. Areas with smaller polling stations and fewer ballot boxes had significantly inflated votes and turnout, again, in favor of Erdoğan. Furthermore, electoral districts with two or fewer ballot boxes were more likely to show large swings in vote shares in favour of Erdoğan from the first to the second round. Based on a statistical model, it is estimated that these shifts account for 342,000 additional ballots (SD 4,900) or 0.64% for Erdoğan, which is lower than the 4.36% margin by which Erdoğan was victorious. Our results suggest that Turkish elections continue to be riddled with statistical irregularities, that may be indicative of electoral fraud.

## Introduction

The first round of the 2023 presidential election was held on 14 May 2023 and pitted incumbent President Recep Tayyip Erdoğan of the Justice and Development Party (AKP) against opposition candidate Kemal Kılıçdaroğlu, who led an alliance of six opposition parties. The election was seen as the first real threat to Erdoğan's presidency in a long time, as Turkey was

**Competing interests:** The authors have declared that no competing interests exist.

reeling from a prolonged economic crisis with inflation rates of up to 85% and the aftermath of devastating earthquakes in February 2023 that killed more than 50,000 people, coupled with public outrage at the government's slow response to these crises [1]. Despite polls to the contrary, Erdoğan won by a healthy margin over Kılıçdaroğlu with 49 per cent to 45 per cent of the vote, sending Turkey into a run-off between the two candidates on 28 May. Erdoğan won this second round of the election with a margin of 4.36% (52.18% versus 47.82%) over Kılıçdaroğlu.

While elections in Turkey are generally considered to be free and fair, the playing field is clearly not level. For example, according to Turkish media monitoring, Erdoğan received almost 33 hours of airtime on the main state television channel, compared to 32 minutes for Kılıçdaroğlu [2]. These and other imbalances became possible after Erdoğan transformed Turkey from a parliamentary democracy to an executive presidency following a 2017 referendum in which a reform package narrowly won. That election, however, was marred by allegations of fraud, as unverified videos and reports emerged on social media showing various forms of electoral malpractice, such as ballot stuffing (casting multiple votes for one candidate) and voter coercion (preventing potentially opposing voters from casting their ballots) [3, 4]. Also practices such as handing out cash to supporters have been reported [5].

The emerging field of electoral forensics seeks to diagnose the extent to which a particular type of malpractice may have affected the outcome of an election, in order to identify electoral malpractice in a timely and fully quantitative manner [6]. A disproportionate abundance of round numbers was often the focus of early work in election forensics. The basic principle of these tests is that humans have a particular tendency to favour round numbers, or numbers with certain digits, when producing results. These tendencies are at odds with the statistics of the expected number and digit distributions of fair elections [7, 8], including violations of Benford's Law [9–11]. However, it has been shown that such digit-based tests need to be combined with contextual information such as country-specific risk factors, socio-economic inequalities or ethnic fractionalisation [12].

As a result, there has been a growing interest in forensic electoral testing that attempts to provide "mechanistic" or generative models of the impact of specific electoral malpractices on the expected distributions of votes and turnout across polling stations, as well as on the correlations between votes and turnout [6, 13–18]. The basic rationale of such approaches is to consider elections as large-scale natural experiments in which a population is divided into a large number of electoral units in which each registered voter makes the decision to (i) cast a valid ballot or not, and (ii) vote for a particular candidate. The large number of electoral units in most countries means that certain statistical regularities can be expected to hold. Election forensics then tests whether deviations from these regularities are consistent with specific types of fraud. Similar principles can be used to apply machine learning models for election forensics [19]. These statistical tools are often complemented by analyses of secondary data, such as exit polls or survey and sampling data [20, 21].

In this study, we concentrate on statistical tests for two electoral malpractices; ballot stuffing and voter coercion. Ballot stuffing is when multiple ballots are cast per voter for the same candidate. This malpractice can result in a correlation between voter turnout and the candidate's vote when done on a large enough scale [15]. Voter coercion or intimidation encompasses malpractices whereby non-supporting voters endure intimidation, such as threats of violence, or undue influence from private individuals, outside groups or state actors [22, 23].

Research indicates that certain demographics or population groups are more vulnerable to voter intimidation, with a sustained history of intimidation targeting ethnic minority groups [24]. Reports from the recent local elections held in 2018 and 2019 highlight that there were infrequently observed irregularities at polling stations. These included inappropriate

behaviours exhibited by party agents, instances of gender-based intimidation among voters, and the difficulty people with disabilities faced in accessing their right to vote [25]. A study conducted in seven African countries has found that voters residing in politically competitive regions are more likely to encounter intimidation tactics compared to those living in incumbent strongholds [26]. Based on reports from the 2011–12 Russian election cycle, it was found that intimidation is particularly likely to occur facilitated by employers in single-company towns with few outside employment options [27]. This form of workplace intimidation was also reported for Romania and Bulgaria, where it was especially common in regions with a small number of large employers [28]. For the 2018 Guatemalan elections, it was discovered that 25% of the rural populace faced intimidation tactics during the campaign, in contrast to only 9% of urban dwellers [29]. The results suggest that political players may be more inclined to use these strategies in remote and isolated areas where it is less probable that the victims will report them, thus lowering the risk of a tarnished reputation [29]. In the 2018 Turkish general election, researchers described a phenomenon known as familial electoral coercion [30]. This practice was prevalent among supporters of the ruling AKP party, particularly those with lower levels of education in the less developed regions of East and South-East Anatolia.

In a forensic analysis of the 2017 election, we confirmed that the election records indeed show specific statistical irregularities that point toward ballot stuffing and voter coercion [31]. Voter manipulation or coercion tends to inflate votes and turnout in smaller or more remote regions, where opponents are easier to identify and irregularities are less likely to be observed or reported [16]. For the constitutional referendum, we found that 11% of areas were potentially affected by ballot stuffing and that removing the affected influences from the data would have turned the overall vote of the referendum from "Yes" to "No". There were also small but significant traces of ballot stuffing. Similar statistical irregularities were also observed in the 2018 presidential and parliamentary elections.

Here we ask whether similar forensic patterns are also present in both rounds of the 2023 presidential election [32]. We test whether statistical fingerprints of ballot stuffing or voter coercion can also be found in the 2023 results. We also seek to clarify the contribution of small and/or remote electoral units to such irregularities. In particular, we develop a statistical test to compare the results at the polling station level for the first and second rounds of the presidential election. With this test, we ask whether boxes in such areas are more likely to show improbably large swings of votes in favour of either candidate from one round to the next.

## Data and methods

We analyse the official and final election results of both rounds of the 2023 presidential election, as provided by the election commission [32]. At the finest level of aggregation available, the data for the first round consists of 191,872 electoral units (here also referred to as "boxes") for which we consider the size of the electorate (number of eligible voters), the number of valid votes and the number of votes for Recep T. Erdoğan. The same variables were extracted from the data for the second round, which contained 192,214 electoral units. Of these, 12 and 18 units respectively were removed because their number of valid votes was zero.

When applying the ballot-stuffing and voter-rigging tests, all electoral units with an electorate of less than one hundred voters are excluded in order to rule out that the results are driven by small number artefacts. This reduces the number of electoral units used to 180,301 and 180,629 respectively. The number of valid votes divided by the size of the electorate, $n_i^{(j)}$, for a ballot box $i$ is called the "turnout", $t_i^{(j)}$, for the first ($j = 1$) or second ($j = 2$) round of the election. The number of votes, $V_i^{c,j}$, for candidate $c$ divided by the valid votes is called the "vote share", $v_i^{(c,j)}$.

The election data includes multiple administrative levels for each ballot box, namely 81 provinces, 948 districts and 28,268 counties. The number of ballot boxes is very unevenly distributed across these districts. In particular, we identify districts that are typically smaller and more remote as those that have only two or fewer ballot boxes.

We apply ballot stuffing [15] and voter coercion [16] to the data as previously reported [31]. In order to investigate whether there is systematic bias in vote shifts, we propose the following test procedure. In the first round, Sinan Oğan also ran and received 5.17% of the vote, after which he supported Erdoğan in the run-off. Therefore, we compare the vote shares of Erdoğan ($c = E$) or Oğan ($c = O$) in round 1 with Erdoğan's shares in round 2. To do this, we compute the "vote shift" at the ballot box level as $\delta v_i = \frac{V_i^{(E,2)}}{t_i^{(2)}} - \frac{V_i^{(E,1)} + V_i^{(O,1)}}{t_i^{(1)}}$. We assume that there is a general population-level trend in how preferences for a candidate may have shifted between the first and second rounds, which can be obtained as the mode of the distribution of $\delta v_i$, denoted as $\bar{\delta v}$. Algorithmically, we estimate $\bar{\delta v}$ using MatLab's kernel density estimation procedure. Our null hypothesis is that the vote shifts at the ballot box level are symmetrically distributed around this mode, i.e. the expectation $E$ for the deviations from $\bar{\delta v}$ is zero, $H_0 : E(\delta v_i \bar{\delta v}) = 0$.

Let $B^{+/-}$ be the set of ballot boxes for which this deviation is greater/smaller than zero, $B^{+/-} = \{i | \delta v_i - \bar{\delta v} > / < 0\}$. A symmetrized vote shift distribution can then be constructed by replacing, for example, $(\delta v_i - \bar{\delta v}) \forall i \in B^+$ by values of $-(\delta v_j - \bar{\delta v})$, where $j$ was randomly sampled from $B^-$. For the case were vote shifts typically favor one candidate, say $E(|\delta v_i \bar{\delta v}|)_{B^+} > E(|\delta v_i \bar{\delta v}|)_{B^-}$ (meaning that the expectation value is taken over all $i$ in $B^{+/-}$), one replaces vales in $B^+$ with values sample from within $B^-$. This gives a model estimate for a symmetrized vote shift distribution from which corrected vote totals for individual candidates can be estimated. By comparing the actual vote tallies with expectations from the model with symmetrized vote shift distributions, one obtains the number of excess votes due to large vote swings.

## Results

In 2018, Erdoğan was elected president with 52.59% of votes, defeating Muharrem İnce who received 30.64% of votes, with an 86.24% turnout. In the first round of the 2023 election, Erdoğan received 49.52% of votes, winning over Kılıçdaroğlu who won 44.88% with an 87.04% turnout. During the second round, Erdoğan obtained 52.18% of votes while Kılıçdaroğlu received 47.82%, with a turnout of 84.15%.

The cumulative percentage of votes for Erdoğan is shown as a function of turnout in Fig 1. For each turnout level (x-axis), the share of votes from boxes with that turnout level or lower is shown on the y-axis. In 2018, the share of votes exceeds the 50% threshold only if we include voting boxes with a turnout of more than 90%. In 2023, we observe a similarly shaped curve with an overall higher turnout (the curve is shifted to the right) but without crossing the 50% threshold in the first round of the election. For the second round we find again a qualitatively similar curve as in 2018 that crosses the 50% threshold at a turnout of around 92%.

### Ballot stuffing test

In Fig 2, we test for the presence of electoral malpractices that lead to vote-turnout correlations (such as ballot stuffing) using so-called electoral fingerprints, i.e., a 2-d histogram of the vote-turnout distribution. The fingerprint for the first round and second round of the 2023 Turkish presidential election is shown in Fig 2(A) and 2(D), respectively. The colour intensity (blue)

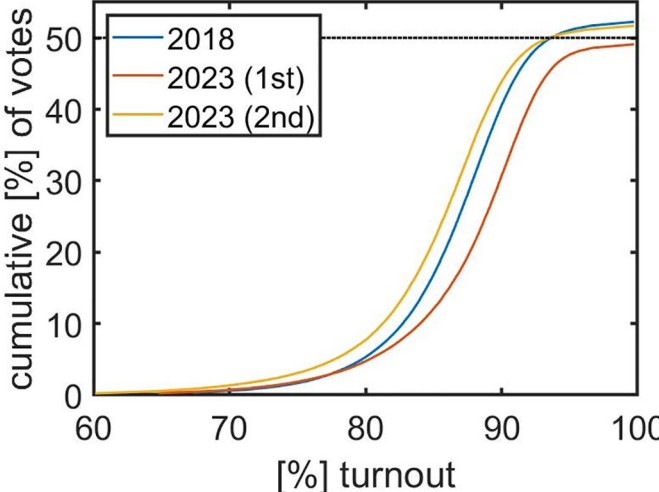

**Fig 1. Votes for Erdoğan as a function of voter turnout for 2018 and 2023.** For a given turnout level, the cumulative vote share of ballot boxes with this or lower turnout is shown. In 2018, a majority of more than 50% is achieved by including boxes with a turnout of more than 90% (blue). In the first round in 2023 (red line) we observe similar characteristics with higher turnout and lower vote shares (below 50%), whereas in the second round (yellow) Erdoğan reached a majority of votes.

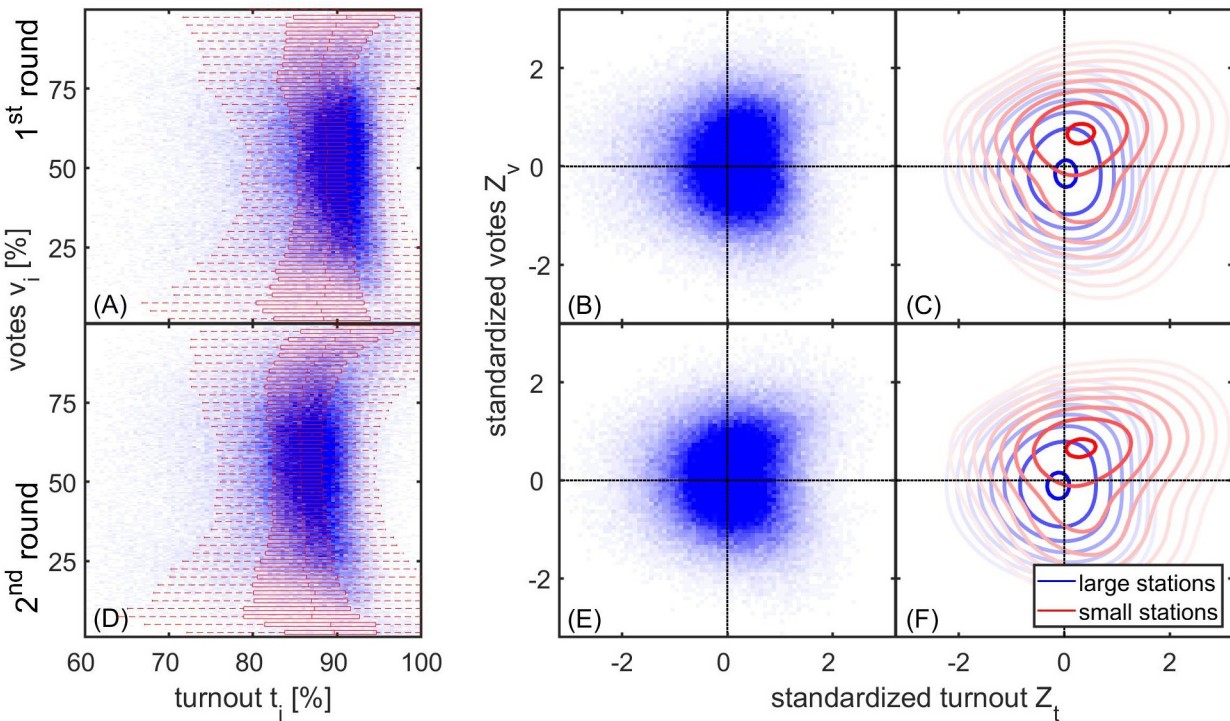

**Fig 2. Forensic electoral fingerprints for the two rounds of the 2023 presidential election.** The fingerprints for (A) first round and (B) second round show joint vote-turnout distributions with color intensity encoding the number of ballot boxes with a given vote (y-axis) and turnout (x-axis). For both elections, we find a visible correlation in the region of high vote and turnout (e.g. more than 80%), which can be associated with ballot stuffing. A box plot (red horizontal boxes) shows the 25th, 50th and 75th percentiles of voter turnout as a function of votes (whiskers indicate the 95% confidence interval). (C) To adjust for regional characteristics, the fingerprints can be adjusted by rescaling the vote and turnout shares by their typical levels in the unit's region, resulting in the standardised fingerprint shown. (D) Traces of voter coercion can be identified by comparing the standardised fingerprints of small (red lines) and large (blue) units, as voter coercion results in their being shifted towards inflated votes and turnout, as observed here. (E,F) The standardised fingerprints of the second round are similar to those of the first round.

indicates the number of boxes with the corresponding percentage of votes (x-axis) and turnout (y-axis), together with a box plot of the distribution of turnout for a given percentage of votes. If there were no non-linear correlations between votes and turnout, the bulk of the distribution in Fig 2(A) and 2(D) should be circular or elliptical symmetric. Malpractices such as ballot stuffing would inflate turnout and simultaneously increase vote shares, breaking the elliptical symmetry in the fingerprints if the number of affected boxes is large enough.

Considering the region of high voting and turnout, in both rounds we see a smearing of the bulk towards inflated votes and turnout, towards values of 100% votes for Erdoğan and 100% turnout. To assess whether such deviations between symmetric and biased fingerprints are statistically significant, we run a parametric test that was proposed previously [31]. This model is designed to test if the observed deviations from the normal distribution in vote and turnout shares can be better explained by a model where ballot stuffing occurs in a given fraction of ballot boxes (fraud parameter $f$). To fit and evaluate this statistical model, we follow a previously described strategy [31] and further restrict the analysis to boxes with a vote and turnout share of more than 25%.

For the first round we find a fraud parameter of $f = 0.024$ with a standard deviation (SD) of 0.019 and for the second round $f = 0.19$ (SD 1.7%). Note that for 2018 we find higher values with $f = 0.085$ (SD 0.039). Therefore, the test suggests that the number of ballot boxes affected by such statistical irregularities has decreased from 2018 to 2023, to a point where the ballot stuffing test does not detect statistically significant effects.

## Voter rigging test

The fingerprints shown in Fig 2(A) and 2(D) may also show deviations from elliptical symmetry due to geographical effects. To account for such effects, it has been suggested to compare the unit with other units in close geographical proximity [16]. Here, we compare the vote and turnout figures of a polling station with the averages observed in other polling stations in the same constituency. We refer to these rescaled vote and turnout shares as standardised votes and turnout, respectively. We call their joint distribution (2D histogram) the "standardised fingerprint". Standardised fingerprints are shown in Fig 2(C) for round 1. For the voter coercion test, we ask whether small and large units have different standardised fingerprints. The underlying hypothesis of this test is that coercion is more likely to occur in smaller units because they are more susceptible to coercion tactics. Reasons for this include that (i) it is easier to identify opponents in smaller units, (ii) fewer eyewitnesses can be expected, and (iii) election observers are less likely to be present. In line with these assumptions, voter manipulation suggests that the standardised fingerprints of small units are biased towards increased voting and turnout compared to larger units.

We use different definitions of "small units". Fig 2(B) shows the standardised fingerprints for small (red) and large (blue) units, where small units are those in the lowest $p = 10$th percentile of all units. It can clearly be seen that the fingerprints for small units are shifted towards the upper right corner, see Fig 2(C), which is consistent with voter manipulation. For the second round, we found almost identical standardised fingerprints, see Fig 2(E) and 2(F).

The magnitude of the displacement between the average standardised votes and the turnout of small and large units depends on the size threshold p and is denoted by $\delta(p)$, see [16, 31] for methodological details. To assess whether this shift is statistically significant, we apply the Jimenez et al. voter rigging test [16]. The idea behind this test is to estimate the expected shifts between small and large units based on a reference set of trustworthy elections, yielding a range of "acceptable shift sizes". We obtain this acceptable region from 21 different reference elections in ten countries, see [16, 31]. For a given election, one can now check whether the

actual observed displacement between small and large units for a size threshold $p$ lies within this region ("accepted region") or not.

The displacement, $\delta(p)$, is shown in Fig 3(A) for the reference set of elections (solid lines), the elections in Russia and Venezuela (dashed line) next to the two rounds in 2023 (solid magenta) and 2018 (solid black) Turkish elections. For small size thresholds, $p$, all Turkish datasets show shifts that are slightly outside the acceptable range. This indicates statistically significant signs of voter manipulation, however, only for a limited region of thresholds. The shifts in the first round in 2023 are slightly smaller than in 2018, whereas in the second round they were slighly larger.

To assess the potential impact of these voter rigging effects in the data, we rank the units by their electorate size in descending order and calculate the vote share over all units with smaller ranks (higher electorate size), see Fig 3(B). In this plot, voter manipulation takes the form of a "hockey stick", i.e., a sharp increase for the smallest units. This signal is also found in Russia and Venezuela, but is absolutely absent in the reference elections, see the insets in Fig 3(B).

To further show the bias in small units in the first and second round in 2023, respectively, we compare the fingerprint observed in all units with an electorate size larger than 100, Fig 4 (A) and 4(B), with the fingerprints of two different definitions of "small". First, we consider

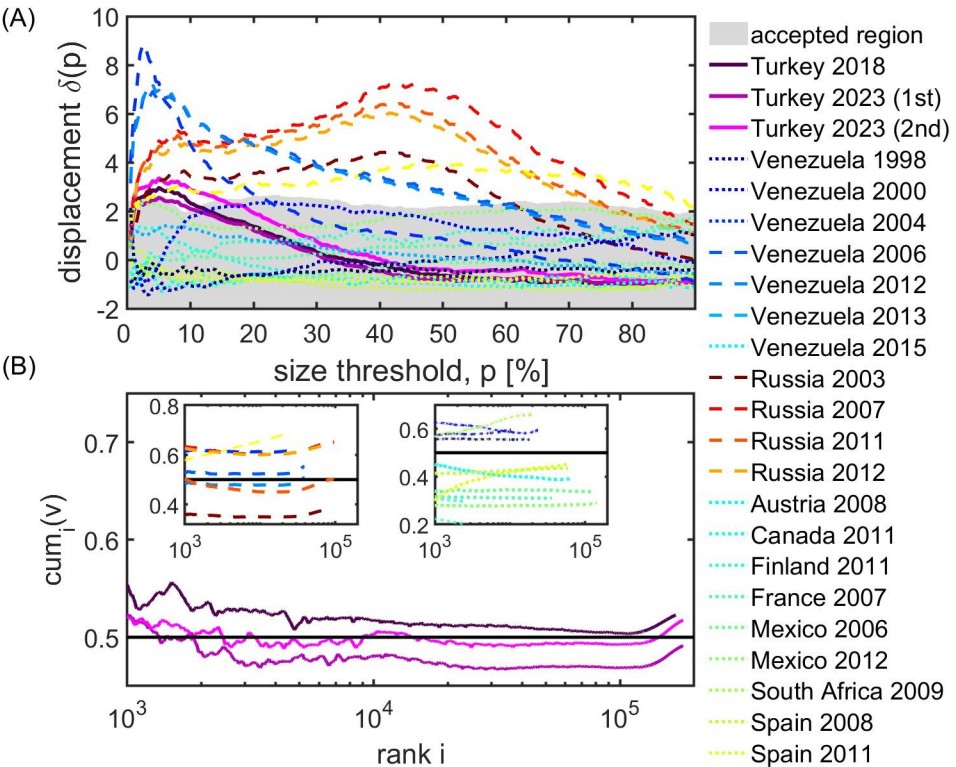

**Fig 3. Results of the statistical test for voter manipulation.** (A) The displacement $\delta(p)$ between small and large units for the first (solid dark magenta line) and second (solid light magenta line) round of the 2023 elections is very narrowly outside the accepted range for a restricted set of size thresholds, similar as it was in 2018 (solid black line). These displacements are much smaller displacements in the Russian or Venezuelan elections (dashed lines); the reference elections are shown as dotted lines. (B) Units in the 2023 and 2018 Turkish elections are ranked according to their electorate size. We show the cumulative vote share, $cum_i(v)$, calculated over all units with a size greater than the given rank. As in 2018, we observe a characteristic "hockey stick" in 2023, meaning that units with high rank (low electorate size) show a clear tendency to favour Erdoğan.

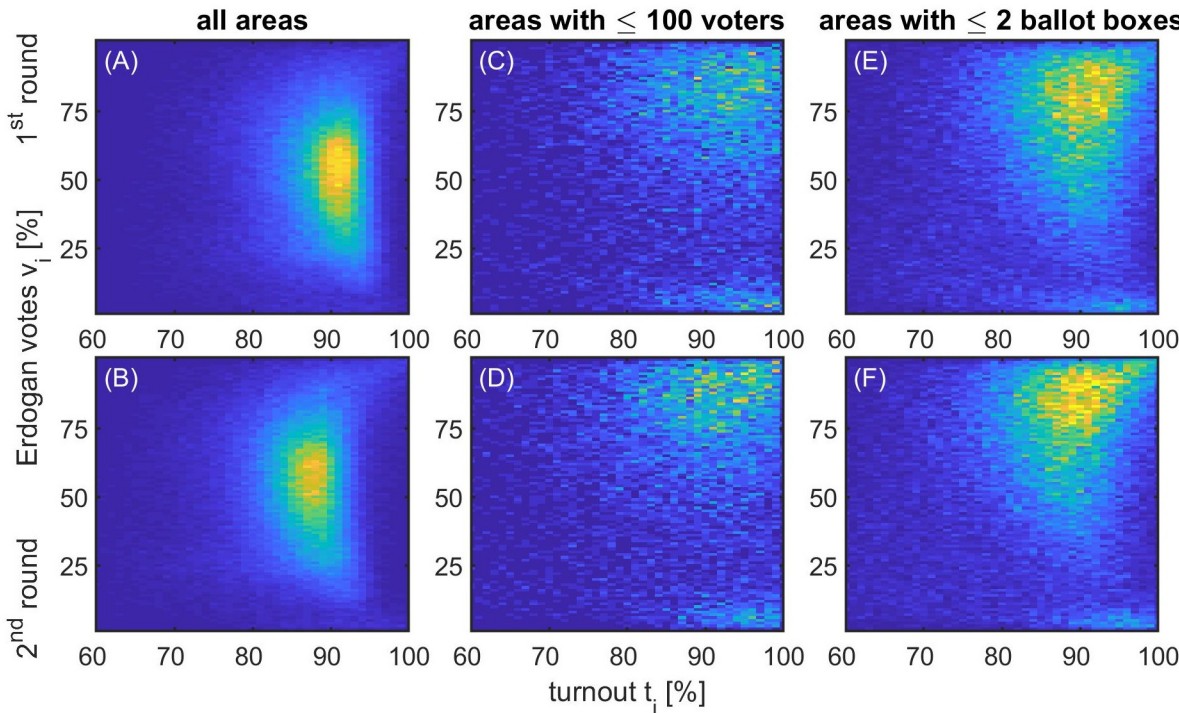

**Fig 4.** The fingerprints in 2023 for small units (C-F) show very different patterns than those with an electorate size of more than 100 (A, B). Considering only areas with an electorate size of less than 100 (C, D), or alternatively ballot boxes from areas with one or two ballot boxes (E, F), results in a bimodal distribution with a large mode in the region with very high turnout and votes for Erdoğan.

only boxes with an electorate of one hundred or less, resulting in 11,689 units, see Fig 4(C) and 4(D). Alternatively, one can consider only boxes from areas with one or two ballot boxes in total, resulting in 38,662 boxes in the first round, see Fig 4(E) and 4(F). The small units show completely different fingerprints when compared to the large ones. They show a bimodal distribution with a larger mode in the high vote-high turnout region and a smaller mode in the low vote-high turnout region.

## Extreme vote shifts

Comparing Fig 4(E) and 4(F), there is a tendency for the high vote/high turnout mode to have shifted further to the upper right of the plot in the second round compared to the first. To systematically compare such effects, we examine the shift of vote shares between Erdoğan or Oğan in the first round and Erdoğan in the second round (see Methods). A scatterplot of these vote shares is shown in Fig 5(A), which shows that in the vast majority of cases the vote shares for Erdoğan (or candidates who later endorsed him) were similar in the first and second rounds; the points cluster around the $x = y$ line. The distribution of vote shifts, $\delta v_i$, in Fig 5(B) shows a clear tendency for large shifts in vote shares from Kılıçdaroğlu in round 1 toward Erdoğan in round 2 to be more common than vice versa (the distribution is skewed to the right, favouring Erdoğan). Fig 5(C) shows the percentage of ballot boxes that are in regions with two or fewer ballot boxes in total for each of the bins in the histogram (bins in the lowest or highest percentile have been combined). Interestingly, larger districts (in the sense of having more than two ballot boxes) have very small vote shifts. Strong positive vote shifts in favour of Erdoğan occur mainly in districts with few ballot boxes.

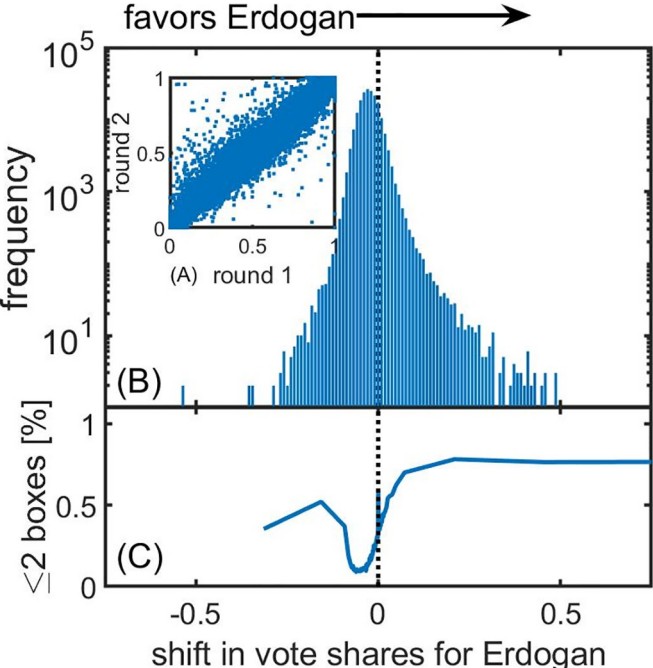

**Fig 5. Statistical analysis of vote shifts from Kılıçdaroğlu to Erdoğan from the first to the second round.** (A) The vote shares for Erdoğan and Oğan in the first round are strongly correlated with Erdoğan's vote share in the second round, although there are some outliers. (B) A histogram of these vote shifts shows that these outliers typically favour Erdoğan in the sense that large shifts from anti to pro Erdoğan are much more common than the other way around. (C) Looking at the proportion of ballot boxes coming from districts with two or fewer ballot boxes as a function of vote shifts, we see that these extreme vote shifts favouring Erdoğan occur predominantly in districts with fewer ballot boxes.

Similar observations apply, to a lesser extent, to vote shifts in favour of Kılıçdaroğlu. To compare the magnitude of these vote shifts, we consider a symmetrized distribution of vote shifts in which vote shifts in favour of Erdoğan follow the same distribution as those in favour of Kılıçdaroğlu (see Methods). Overall, we find that Erdoğan received 342,000 (SD 4,900) excess votes, which corresponds to 0.64% of all valid votes.

In the following, extreme vote shifts are defined as those surpassing the nationwide average vote shift (the mean of the histogram in Fig 5(B)) by five standard deviations or more, equating to 14.5%. If the vote shifts displayed a normal distribution, an event with such a vote shift would occur once every 3.5 million observations. However, we find 475 of the 191,873 ballot boxes with such five sigma events in vote shifts, occurring in 0.25% of all cases. To determine if these events are concentrated in specific areas, we calculated the frequency of these occurrences in each region. We then calculated the p-value under the null hypothesis that the vote shifts follow a binomial distribution with an expected success probability of 0.25%. After Bonferroni correction for multiple testing, it has been observed that 38 districts have significantly overrepresented extreme vote shifts, as shown in Table 1. The frequencies range from 2.2% in the Siirt Merkez district in the Siirt province to 13% in Hilvan in the Şanlıurfa province. It is noteworthy that all of these districts are located in the East or South East of Turkey, with Siirt and Şanlıurfa having the most frequent occurrences (five each).

Note that in the analysis above, we combined the vote tallies of Erdoğan and Oğan in round 1. As Oğan endorsed Erdoğan after round 1, we assume that his votes were also shifted towards Erdoğan. However, in a robustness test, we refrain from making this assumption and calculate

**Table 1. Districts with significant extreme vote shifts.** Overall, 0.25% of ballot boxes were affected by vote shits of more than 14.5% (which is more than five standard deviations higher than the typical vote swings). The table gives the 38 districts in which ballot boxes with such extreme vote shifts occurred significantly more often than expected (after Bonferroni adjustment for mutliple testing).

| district | province | % ballot boxes with extreme swings |
|---|---|---:|
| SAMSAT | ADIYAMAN | 8.3% |
| PATNOS | AĞRI | 5.9% |
| TUTAK | AĞRI | 6.1% |
| BEŞİRİ | BATMAN | 8.0% |
| GERCÜŞ | BATMAN | 9.3% |
| HASANKEYF | BATMAN | 7.1% |
| ADİLCEVAZ | BİTLİS | 4.1% |
| BİTLİS MERKEZ | BİTLİS | 4.3% |
| ÇERMİK | DİYARBAKIR | 3.8% |
| ÇINAR | DİYARBAKIR | 3.6% |
| KARAYAZI | ERZURUM | 5.9% |
| KÖPRÜKÖY | ERZURUM | 7.8% |
| TEKMAN | ERZURUM | 4.7% |
| ŞEMDİNLİ | HAKKARİ | 4.8% |
| TUZLUCA | IĞDIR | 7.5% |
| KAĞIZMAN | KARS | 4.3% |
| SELİM | KARS | 4.2% |
| DERİK | MARDİN | 11% |
| KIZILTEPE | MARDİN | 4.3% |
| MAZIDAĞI | MARDİN | 8.0% |
| MİDYAT | MARDİN | 3.5% |
| NUSAYBİN | MARDİN | 2.9% |
| MALAZGİRT | MUŞ | 5.1% |
| MUŞ MERKEZ | MUŞ | 2.3% |
| ERUH | SİİRT | 5.5% |
| KURTALAN | SİİRT | 3.6% |
| PERVARİ | SİİRT | 4.5% |
| SİİRT MERKEZ | SİİRT | 2.2% |
| ŞİRVAN | SİİRT | 6.8% |
| GÜRPINAR | VAN | 6.8% |
| ÇALDIRAN | VAN | 3.1% |
| BİRECİK | ŞANLIURFA | 3.0% |
| HİLVAN | ŞANLIURFA | 13% |
| SURUÇ | ŞANLIURFA | 3.7% |
| SİVEREK | ŞANLIURFA | 4.3% |
| VİRANŞEHİR | ŞANLIURFA | 5.2% |
| BEYTÜŞŞEBAP | ŞIRNAK | 6.1% |
| İDİL | ŞIRNAK | 5.7% |

the vote shifts using only Erdoğan's votes from round 1. We identified 516 ballot boxes with five sigma events in their vote shifts, corresponding to shifts of 17.3% or more. In a further test of robustness, we calculated vote shifts using exclusively the results for Kılıçdaroğlu from rounds 1 and 2. As a result, we identified 417 districts with significant vote shifts of 14.9% or more away from Kılıçdaroğlu. Finally, we excluded all ballot boxes from the analysis in which

Oğan and İnce together received more than 10% of the votes in round 1. The number of ballot boxes with significant vote shifts of more than 14.0% changes then to 498. If we decrease this threshold to 5% of votes for Oğan and İnce, we already remove more than half of the ballot boxes. However, we still find 342 ballot boxes with significant vote shifts of 15.8% or more. Our findings remain robust under different strategies of accounting for the round 1 votes of candidates other than Erdoğan and Kılıçdaroğlu.

## Discussion

An electoral forensic analysis of the first and second round of the 2023 presidential election in Turkey identifies statistical irregularities similar to those observed in the 2018 election and the 2017 constitutional referendum. However, the estimated magnitude of these irregularities has decreased in 2023 compared to the 2018 presidential election. For 2023, we observe trends in turnout inflation, as would be expected in the presence of electoral malpractices such as ballot stuffing. However, the percentage of electoral units potentially affected by these distortions has fallen to 2.4% (SD 1.9%) in the first round and to 1.9% (SD 1.7%), making the results statistically insignificant. It is probable that Erdoğan would have emerged victorious in most of these units even without inflated votes and turnouts, so the impact of these irregularities on the vote counts is anticipated to be significantly less than 2.4% and 1.9%, respectively. In both rounds in 2023, we also observe a tendency for areas with small electorates to show different voting and turnout patterns compared to other regions. Such biases are consistent with the presence of voter coercion or intimidation techniques, to which smaller and more remote electoral units are more susceptible. The effect size of these deviations is statistically significant only by a small margin and for a limited range in the parameter space (meaning these effects vanish when increasing the size threshold of what constitutes "small" unit above the 20th percentile).

The causes behind the reduction of irregularities in 2023 compared to 2017/18 are not entirely clear. As far as we know, up until now, no other study has exhaustively and comprehensively examined how the frequency of electoral malpractices has recently altered in Turkey. Instances of malpractices such as voter intimidation often come with reputational costs when they are reported, hence their prevalence in remote and secluded regions. [28, 29] Given the rise in Erdoğan's influence following the 2017 constitutional referendum, there may have been less expensive options available to the incumbent candidate. For instance, Ekrem İmamoğlu, the Mayor of Istanbul, was viewed as a potential opposition candidate for the presidency based on the magnitude of his success in the 2019 mayoral election. However, in 2022 he faced a trial for insulting electoral officials and subsequently received a ban from politics (although the ban has yet to be implemented). The verdict received significant criticism for its political motivation, with Human Rights Watch referring to it as a "travesty of justice and an attack on the democratic process" [33]. Another possible explanation for the reduction might be increased efforts for ballot box security by the opposition after experiences from the 2017 and 2018 elections.

We found, however, suprising irregularities when considering ballot boxes in areas with two or fewer ballot boxes in total, which typically hints at remote areas. There are roughly 40,000 such boxes (out of 190,000 in total) that show completely different trends in their vote–turnout distribution when compared to the other areas. In such remote areas the vote–turnout distribution becomes bimodal with high vote/high turnout bulks for both candidates (Erdoğan and Kılıçdaroğlu, respectively), though the Erdoğan mode contains much more ballot boxes compared to the Kılıçdaroğlu mode. What is more, comparing the first and the second round of the election, a much higher fraction of such boxes has flipped from the Kılıçdaroğlu to the Erdoğan mode compared to the other way around. This high number of districts (around

90%) that flipped from one candidate to the other at very high turnout levels in the span of no more than two weeks is certainly surprising. Correcting the election results for such suprising vote shifts would reduce Erdoğan's vote tally by about 342,000 votes or 0.64% of valid votes.

Shortly after the first round of the presidential election, held alongside the parliamentary election, the hashtag "#OylarNewsCount" ("#oylaryenidensayılsın") trended on Twitter [34]. Citizens used this hashtag to share comparisons between recorded votes on a ballot-box level in the parliamentary election reported by the CHP and the official election commission, YSK [32]. Apparently, these comparisons revealed that votes for the CHP were systematically shifted to other parties or cancelled altogether when they were entered into the official reporting system. These reports mostly originated from the Eastern and South Eastern regions. Our analysis of the extreme vote shifts indicates that we cannot rule out the possibility that this type of manipulation occurred in the presidential election.

Specific forms of voter intimidation, such as economic [28] or familial coercion [30], should be expected toexert similar effects on both rounds of the election, and are therefore unlikely to result in extreme shifts in votes. However, our voter rigging test using standardized election fingerprints should identify this form of voter suppression, and we do in fact find statistical evidence to support this.

In the analysis of vote shift, it is necessary to consider the possibility that these shifts may be partially influenced by candidates who only competed in the first round and not in the run-off election. Sinan Oğan and Muharrem İnce received 5.17% and 0.43% of the votes, respectively, in the first round. To investigate whether the substantial shifts in votes are caused by the movement of those votes to Erdoğan and Kılıçdaroğlu, several robustness tests were conducted. Firstly, we examined several options in which the votes for Oğan are allocated to Erdoğan (whom Oğan later supported) or not. Secondly, we omitted ballot boxes with a substantial proportion of votes for Oğan and İnce entirely from the vote shift analysis. In both sets of tests, we identified similar quantities of ballot boxes with significant vote shifts, indicating that these shifts are not influenced by votes for Oğan and İnce. Note that our other tests (ballot stuffing and voter rigging) are not affected by this issue. They compare different regions within the same election rather than the same region across different elections in the vote shift analysis.

When interpreting the results of forensic voting tests, several limitations must be borne in mind. First and foremost, none of these tests can provide incontrovertible evidence of electoral fraud; they provide correlation, not causation. Although most of the tests make adjustments for regional characteristics and rule out the possibility that these correlations are spurious artefacts from small districts, it is not possible to control for all potential confounders. More specifically, the voter rigging test exclusively examines ballot boxes from a particular district against other ballot boxes within the same district. Therefore, the test accounts for discrepancies in levels of urbanisation and other socioeconomic features among approximately 1,000 Turkish districts. Nevertheless, the test does not account for such differences that exist within the confines of the same district. Further, a positive result in such a test indicates that the data are compatible with certain types of fraud (ballot stuffing, etc.) and typically gives a quantitative estimate of how many regions might have been affected by such malpractice. Other nonfraudulent influences can never be completely ruled out. In order to understand whether the observed irregularities may be due to such non-fraudulent phenomena, such as heterogeneous voter mobilisation through strategic voting, forensic tests must always be evaluated in conjunction with external information [35, 36]. Conversely, even if the tests give a negative result, one cannot rule out the presence of other types of fraud, which would require a different test.

Taken together, these results suggest that the presence of certain types of electoral malpractice in both rounds of the 2023 presidential election cannot be ruled out. However, these malpractices appear to have been less frequent in 2023 than in 2018, and less decisive in swinging

the vote one way or the other. There is a notion that Turkish elections have become "free and unfair" because Turkey's political playing field is known to be massively tilted in Erdoğan's favour [37]. It has been described that Erdoğan or his supporters have autocratised the media, the judiciary, civil society and academia [38, 39]. Our analysis suggests that it may be more appropriate to describe Turkish elections as "mostly free and unfair", as consistent trends of small but discernible electoral irregularities can be consistently found by electoral forensic tools. While these statistical irregularities were not large enough to determine the outcome in 2023 on their own, they certainly tilt Turkey's political playing field even further towards an electoral autocracy.

## Supporting information

**S1 Dataset. Election data.** For each analyzed election, results are provided as separate sheets within a single XLSX files. Each row corresponds to an electoral unit (ballot box for elections in Turkey). The columns correspond to the size of the vote eligible population, $n_i$, thurnout, the votes for the winner, $V_i$. and an index labelling the administrative region.
(XLSX)

## Author Contributions

**Conceptualization:** Peter Klimek, Ahmet Aykaç, Stefan Thurner.

**Data curation:** Ahmet Aykaç.

**Formal analysis:** Peter Klimek.

**Investigation:** Peter Klimek.

**Methodology:** Peter Klimek.

**Writing – original draft:** Peter Klimek.

**Writing – review & editing:** Peter Klimek, Ahmet Aykaç, Stefan Thurner.

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
