## [Decision Letter · Decision Letter 0]

21 Aug 2023

PONE-D-23-19758Forensic analysis of the Turkey 2023 presidential election reveals extreme vote swings in remote areasPLOS ONE

Dear Dr. Klimek,

Thank you for submitting your manuscript to PLOS ONE. After careful consideration, we feel that it has merit but does not fully meet PLOS ONE’s publication criteria as it currently stands. Therefore, we invite you to submit a revised version of the manuscript that addresses the points raised during the review process.

We look forward to receiving your revised manuscript.

Kind regards,

Ali B. Mahmoud, Ph.D.

Academic Editor

PLOS ONE

Reviewers' comments:

Reviewer's Responses to Questions

**Comments to the Author**

1. Is the manuscript technically sound, and do the data support the conclusions?

Reviewer #1: Yes

Reviewer #2: Yes

Reviewer #3: Yes

2. Has the statistical analysis been performed appropriately and rigorously? 

Reviewer #1: Yes

Reviewer #2: Yes

Reviewer #3: Yes

3. Have the authors made all data underlying the findings in their manuscript fully available?

Reviewer #1: Yes

Reviewer #2: Yes

Reviewer #3: Yes

4. Is the manuscript presented in an intelligible fashion and written in standard English?

Reviewer #1: Yes

Reviewer #2: Yes

Reviewer #3: Yes

5. Review Comments to the Author

Reviewer #1: PONE-D-23-19758

Forensic analysis of the Turkey 2023 presidential election reveals extreme vote swings in remote areas

This is another in a series of papers by the lead author on the forensic analysis of Turkish elections. The authors’ previous research has shown significant evidence of voting irregularities between the first and second rounds of the 2018 and now 2023 Turkish Presidential elections. Previous work has built on Benford’s Law and their own statistical tests of ‘irregular vote shifts’ between rounds of voting. In this paper, the authors offer, test and appear to confirm a novel twist and explanation for how these irregular shifts in votes occur. I would support publication of the paper with some modest revisions. Below I detail what might be addressed in a revised manuscript.

Previous forensic analysis (see Mebane) of voting irregularities has been short on the mechanics with which voter fraud (i.e., ballot stuffing) and coercion occur or at least go undetected at their origin i.e., polling places. In this paper, the authors suggest that small voting locations with fewer ballot boxes are more susceptible to ballot stuffing and voter coercion. Their findings appear to support this small voting place effect on vote shifts that occurred in the 2023 Turkish Presidential election. I am not questioning the authors’ findings, but rather the brevity of their explanation and how this phenomenon is practiced.

The authors suggest that coercion is easier because opponents of the Erdogen are easier to identify. How so? How might supporters of one candidate know how others might vote? Here the authors need to provide a bit more information/description and possibly citations to other supporting research (see literature on polling place voter intimidation e.g., James and Clark special issue of Policy Studies 2020). In the U.S. setting, voter intimidation and coercion follows the race and ethnicity of the voter. Even age and gender can become the basis of coercing voters. How does this operate in Turkish elections, especially in smaller/two or few ballot box locales?

Similarly, ballot stuffing is explained in small polling stations as a function of fewer eyewitnesses and election observers. Is this true? Is there any evidence to support these assumptions? Are there any newspaper accounts or data from official election observers of either party that would show fewer observers et al at locations with fewer polling boxes? However, intuitive some supporting facts is needed.

The 2013 Turkish case had a third place candidate endorse Erdogan which might have affected the turnout, vote shares and relationship between vote shares and turnout used to explain the vote shift between rounds of voting. The authors’ acknowledge this condition, and I assume account for it in their test/estimate of expected shifts between small and large units (see Klimek 2017;2018). I would like a further discussion of how this test accommodated this endorsement of Erdogan or at least more detail reference to the authors’ earlier work. For example, is it customary in recent Turkish elections for candidates that do not make the second round to endorse one of the candidates in the second round? Alternatively, is this issue addressed by using estimates from other countries?

The authors suggest that the degree of voting irregularity specifically irregular voting shifting declined between the 2018 and 2023 election. Though this might be beyond the scope of the paper, why did this occur? At least some thoughts on this finding and what avenues of research are suggested by this finding are worthy of inclusion in a modest revision.

Reviewer #2: This is a well designed, well executed, and timely research piece -- timely because this election took place recently. It takes into account, and builds on, the evolving scholarship to perform forensic analyses of election results. The authors' data is both appropriate and sufficient for these tests. The authors employ two pertinent benchmarks, namely, the results of previous presidential elections in Turkey and the results of elections in other countries, including Russia and Venezuela where the a priori suspicion of electoral fraud is significant. The statistical analysis is persuasive.

As a reader, I would find it helpful if the authors were to include, very briefly, the following descriptive information:

1) Actual vote totals for each of the pertinent elections in Turkey, namely, 2023 and preceding as referred to in the mss.

2) Claims from the opposition regarding electoral fraud as well as responses from Erdogan's team to those allegations. Some assessment whether these qualitative verbal claims are approximately identical in tone, salience, and magnitude in the 2023 and previous presidential elections. That is, does the analysis of the authors -- less fraud in 2023 --match, or not, the verbal disputation between government and opposition. All of this should not take more than a couple of paragraphs.

Reviewer #3: This article answers an important question for the observers of Turkish politics but also for scholars who study electoral autocracies in general. I find the current version of the article publishable after a round of minor revisions. I will list some considerations that would further contextualize the statistical findings and offer a more coherent theoretical framework.

At the end of the article, the authors use the term illiberal democracy. While that might be true for Turkey in the early 2010s, scholars currently describe it as one of the three: hybrid regime, electoral autocracy, or competitive authoritarian regime. The statistical findings also suggest elections that are mostly free and disfavor the opposition. So, Turkey is clearly not a democracy but a type of authoritarian regime that keeps elections mostly free but does not allow the opposition to compete on equal ground.

The authors should include at least a sentence on the Turkish opposition's attempts to ensure ballot box security. The increased attention to this issue in the runoff might explain some of the findings here. The diagnosis is important but there have been some attempts to prevent it.

How does the rural-urban polarization factor into the findings? In general, the article should pay attention to the role of polarization in Turkish politics. Turkey is considered one of the most polarized countries in the world. It is definitely more polarized than the reference elections you picked, perhaps with the exception of Venezuela. It could be possible that Erdoğan did really well in heavily rural areas and small communities. What you are finding might be polarization, not fraud. Please address this potential criticism.

Also, what is the role of the semi-feudal (aşiret) structure in some of the rural areas? How many of the questionable ballot boxes were in Urfa for example? Is there a regional aspect of irregularities other than the size of the ballot box?

It is true that Sinan Oğan backed Erdoğan in the runoff but the Victory Party ended up supporting Kılıçdaroğlu. In the runoff, Kılıçdaroğlu narrowed the margin with Erdoğan.

Large vote shifts from Kılıçdaroğlu to Erdoğan require a more detailed explanation. Could the authors provide some examples from individual ballot boxes that had a big change? This would make the article more accessible to general readers in Turkish politics as well.

The authors say that 2.4 and 1.9 percent of the electoral units were affected by distortions. This could be true but does not invalidate all the votes in that ballot box? Those places would probably vote for Erdoğan anyway, with a smaller margin and turnout.

In the abstract, the authors should indicate that 0.64 percent is not enough to swing the election from Erdoğan to the opposition. Erdoğan would still win the elections without the irregularities.

The authors should spend more time on the declining level of electoral irregularities between 2018 and 2023. Why would this be?

If the authors are going to use Turkish characters, they should spell the opposition candidate's last name as Kılıçdaroğlu.

Overall, I would like to see more connections between the statistical findings and the Turkish context. The current version gives us a general view but needs "contextual information and country risk factors."

6. PLOS authors have the option to publish the peer review history of their article (what does this mean?). If published, this will include your full peer review and any attached files.

Reviewer #1: No

Reviewer #2: No

Reviewer #3: No

---

## [Author Response · Author response to Decision Letter 0]

12 Sep 2023

We thank the editors for the swift review process so far and the reviewers for their very helpful and constructive remarks. We have taken each of their points into account and changed the manuscript accordingly. In particular, we have substantially extended the literature review on voter intimidation, added an analysis where we identify hotspot districts for extreme vote shifts, added robustness tests and extended the discussion to reflect the additional analyses. In the following, we give a point-by-point response on how the manuscript was changed.

REVIEWER#1:

Forensic analysis of the Turkey 2023 presidential election reveals extreme vote swings in remote areas

This is another in a series of papers by the lead author on the forensic analysis of Turkish elections. The authors’ previous research has shown significant evidence of voting irregularities between the first and second rounds of the 2018 and now 2023 Turkish Presidential elections. Previous work has built on Benford’s Law and their own statistical tests of ‘irregular vote shifts’ between rounds of voting. In this paper, the authors offer, test and appear to confirm a novel twist and explanation for how these irregular shifts in votes occur. I would support publication of the paper with some modest revisions. Below I detail what might be addressed in a revised manuscript.

Previous forensic analysis (see Mebane) of voting irregularities has been short on the mechanics with which voter fraud (i.e., ballot stuffing) and coercion occur or at least go undetected at their origin i.e., polling places. In this paper, the authors suggest that small voting locations with fewer ballot boxes are more susceptible to ballot stuffing and voter coercion. Their findings appear to support this small voting place effect on vote shifts that occurred in the 2023 Turkish Presidential election. I am not questioning the authors’ findings, but rather the brevity of their explanation and how this phenomenon is practiced.

The authors suggest that coercion is easier because opponents of the Erdogen are easier to identify. How so? How might supporters of one candidate know how others might vote? Here the authors need to provide a bit more information/description and possibly citations to other supporting research (see literature on polling place voter intimidation e.g., James and Clark special issue of Policy Studies 2020). In the U.S. setting, voter intimidation and coercion follows the race and ethnicity of the voter. Even age and gender can become the basis of coercing voters. How does this operate in Turkish elections, especially in smaller/two or few ballot box locales?

OUR RESPONSE:

We have substantially extended the literature review of voter intimidation in the introduction and summarizing how voter intimidation was found to occur across different elections and regions. We identified previous work on voter intimidation in the 2017/2018 Turkish elections, which found that this malpractice was particularly likely to occur in the Eastern and Southeastern Regions of Turkey among supporters of the AKP with lower levels of educations in less developed regions and through familial voter coercion (see Toros and Birch, Democratization 2019). Interestingly, our new regionalized analysis yields similar results for 2023 regarding the hotspot regions, though based on our analysis we cannot make any statements whether it was familial or, say, employer electoral coercion.

REVIEWER 1:

Similarly, ballot stuffing is explained in small polling stations as a function of fewer eyewitnesses and election observers. Is this true? Is there any evidence to support these assumptions? Are there any newspaper accounts or data from official election observers of either party that would show fewer observers et al at locations with fewer polling boxes? However, intuitive some supporting facts is needed.

RESPONSE:

The introduction now summarizes some relevant empirical findings in this regard. For instance, in Guatemala it was found that 25% of the rural population was confronted with intimidation tactics compared to 9% of the urban population. Further, studies from Russia, Romania and Bulgaria describe the occurrence of “economic electoral coercion” that appears to be mediated by employers and that was found to be particularly frequent in single-company small towns with few employment opportunities outside this single company. Together with the findings by Toros and Birch summarized above, this reveals a clear pattern in which voter intimidation is more likely to be found in isolated and remote areas in which there is a lower risk of reputational costs for actors responsible for the coercion.

REVIWER 1:

The 2013 Turkish case had a third place candidate endorse Erdogan which might have affected the turnout, vote shares and relationship between vote shares and turnout used to explain the vote shift between rounds of voting. The authors’ acknowledge this condition, and I assume account for it in their test/estimate of expected shifts between small and large units (see Klimek 2017;2018). I would like a further discussion of how this test accommodated this endorsement of Erdogan or at least more detail reference to the authors’ earlier work. For example, is it customary in recent Turkish elections for candidates that do not make the second round to endorse one of the candidates in the second round? Alternatively, is this issue addressed by using estimates from other countries?

RESPONSE:

We have added two different robustness tests for how votes from other candidates in round 1 can be accounted for. We found the main results (i.e., number of boxes with extreme vote swings and their magnitude) to be reasonably robust across these different definitions. This is now also stated in the discussion. Note that the other tests are not affected by this. In all other tests, each election is analysed separately and the observables for individual candidates and elections do not play a role (that is, the algorithms typically look at the fraction of votes for a specific candidate and do not depend on the number of other candidates in the race). This issue becomes only prevalent in the vote shift analysis, as where differences in votes between two elections are compared on the ballot box level. This, too, is discussed now in the manuscript.

REVIEWER 1:

The authors suggest that the degree of voting irregularity specifically irregular voting shifting declined between the 2018 and 2023 election. Though this might be beyond the scope of the paper, why did this occur? At least some thoughts on this finding and what avenues of research are suggested by this finding are worthy of inclusion in a modest revision.

RESPONSE:

We have added a paragraph in the discussion on this. To the best of our knowledge, there is no study that has analysed this issue comprehensively to date. In the discussion, we now speculate that this might be due to the reputational costs associated with outright voter intimidation and ballot stuffing and that with the increased power after the 2017 constitutional referendum, less costly options might have opened up. More concretely, we refer to the verdict against Ekrem Imamoglu, the Mayor of Istanbul, who was banned from politics in a verdict that was heavily criticised as being politically motivated. Another potential explanation are increased efforts for ballot box security by the opposition. However, we can only speculate at this point behind the causes of the observed reduction.

REVIEWER 2: 

This is a well designed, well executed, and timely research piece -- timely because this election took place recently. It takes into account, and builds on, the evolving scholarship to perform forensic analyses of election results. The authors' data is both appropriate and sufficient for these tests. The authors employ two pertinent benchmarks, namely, the results of previous presidential elections in Turkey and the results of elections in other countries, including Russia and Venezuela where the a priori suspicion of electoral fraud is significant. The statistical analysis is persuasive.

As a reader, I would find it helpful if the authors were to include, very briefly, the following descriptive information:

1) Actual vote totals for each of the pertinent elections in Turkey, namely, 2023 and preceding as referred to in the mss.

RESPONSE:

Thank you for this suggestion, we have added these numbers at the beginning of the results section.

REVIEWER 2: 

2) Claims from the opposition regarding electoral fraud as well as responses from Erdogan's team to those allegations. Some assessment whether these qualitative verbal claims are approximately identical in tone, salience, and magnitude in the 2023 and previous presidential elections. That is, does the analysis of the authors -- less fraud in 2023 --match, or not, the verbal disputation between government and opposition. All of this should not take more than a couple of paragraphs.

RESPONSE:

We have substantially expanded the discussion in this direction. We are not aware of any comprehensive studies on whether allegations have become more or less frequent in Turkey and state this now. We speculate on potential causes for our observations, namely (i) the use of reputationally less costly strategies, such as the allegedly politically motivated verdict against the Mayor of Istanbul and (ii) increased ballot box security efforts by the opposition. Finally, we relate our analysis of extreme vote shifts to social media reports of vote shifts occurring as the counts were entered into the official reporting system of the YSK. 

REVIEWER 3: 

Reviewer #3: This article answers an important question for the observers of Turkish politics but also for scholars who study electoral autocracies in general. I find the current version of the article publishable after a round of minor revisions. I will list some considerations that would further contextualize the statistical findings and offer a more coherent theoretical framework.

At the end of the article, the authors use the term illiberal democracy. While that might be true for Turkey in the early 2010s, scholars currently describe it as one of the three: hybrid regime, electoral autocracy, or competitive authoritarian regime. The statistical findings also suggest elections that are mostly free and disfavor the opposition. So, Turkey is clearly not a democracy but a type of authoritarian regime that keeps elections mostly free but does not allow the opposition to compete on equal ground.

RESPONSE:

Thank you for pointing this out, we changed the term to “electoral autocracy”.

REVIEWER 3: 

The authors should include at least a sentence on the Turkish opposition's attempts to ensure ballot box security. The increased attention to this issue in the runoff might explain some of the findings here. The diagnosis is important but there have been some attempts to prevent it.

RESPONSE:

Excellent suggestion, we have added this as a possible explanation.

REVIEWER 3: 

How does the rural-urban polarization factor into the findings? In general, the article should pay attention to the role of polarization in Turkish politics. Turkey is considered one of the most polarized countries in the world. It is definitely more polarized than the reference elections you picked, perhaps with the exception of Venezuela. It could be possible that Erdoğan did really well in heavily rural areas and small communities. What you are finding might be polarization, not fraud. Please address this potential criticism.

RESPONSE:

The voter rigging test compares ballot boxes in one district to the “average ballot box” in the same district. Hence, boxes in, say, a district of Istanbul are only compared with other boxes in the same district of Istanbul. Unless there is strong rural-urban polarization within the same district, our results are unlikely to be caused by this form of polarization. We discuss this now explicitly as a limitation.

REVIEWER 3: 

Also, what is the role of the semi-feudal (aşiret) structure in some of the rural areas? How many of the questionable ballot boxes were in Urfa for example? Is there a regional aspect of irregularities other than the size of the ballot box?

RESPONSE:

We have added an analysis to reveal regional irregularities in extreme vote shifts. We find that they concentrate in Eastern and Southeaster regions, mostly in the provinces of Siirt and Şanlıurfa. We also give the districts within these provinces that are most affected by the irregularities.

It is true that Sinan Oğan backed Erdoğan in the runoff but the Victory Party ended up supporting Kılıçdaroğlu. In the runoff, Kılıçdaroğlu narrowed the margin with Erdoğan.

REVIEWER 3: 

Large vote shifts from Kılıçdaroğlu to Erdoğan require a more detailed explanation. Could the authors provide some examples from individual ballot boxes that had a big change? This would make the article more accessible to general readers in Turkish politics as well.

RESPONSE:

We have added several robustness tests for how to handle the votes for Oğan in the vote shift analysis and found our results to be robust with respect to all of these tests. Hence, they are unlikely to be the reason for the observed vote shifts (i.e., votes from Oğan in round 1 shifting to another candidate in round 2). We discuss our results in conjunction with reports of alleged vote shifts that have been reported in the parliamentary elections, where discrepancies between the ballot box results recorded on paper and the official numbers entered in the reporting system appeared. We deliberately chose not the give examples for individual ballot boxes but rather for districts. The reason is that we cannot say for one specific ballot box – on statistical grounds – that these counts are significant statistical anomalies. We can say, however, that boxes with high vote shifts tend to cluster in certain districts and provinces. Therefore we think that this way of reporting the results is more appropriate than singling out individual boxes for which we cannot make clear statements.

REVIEWER 3: 

The authors say that 2.4 and 1.9 percent of the electoral units were affected by distortions. This could be true but does not invalidate all the votes in that ballot box? Those places would probably vote for Erdoğan anyway, with a smaller margin and turnout.

RESPONSE:

Yes, indeed, we clarified this in the discussion.

REVIEWER 3: 

In the abstract, the authors should indicate that 0.64 percent is not enough to swing the election from Erdoğan to the opposition. Erdoğan would still win the elections without the irregularities.

RESPONSE.

We added the margin by which Erdogan won (4.36%) as a reference to the abstract.

REVIEWER 3: 

The authors should spend more time on the declining level of electoral irregularities between 2018 and 2023. Why would this be?

RESPONSE:

We have added a part in the discussion to discuss this. In brief, the two most likely potential explanations for us are that (i) efforts for increased ballot box security on the side of the opposition paid off and (ii) actors avoided such malpractices due to their reputational costs and chose to employ cheaper tactics. More specifically, we refer to the allegedly politically motivated verdict against the Mayor of Istanbul who, until then, was one of the most promising potential opposition candidates due to his victory in the 2019 mayoral election.

REVIEWER 3: 

If the authors are going to use Turkish characters, they should spell the opposition candidate's last name as Kılıçdaroğlu.

RESPONSE:

Thank you for making us aware of this, we corrected the spelling.

REVIEWER 3: 

Overall, I would like to see more connections between the statistical findings and the Turkish context. The current version gives us a general view but needs "contextual information and country risk factors."

RESPONSE:

We hope that the changes described above provide the desired connections between statistical findings and Turkish context.

---

## [Decision Letter · Decision Letter 1]

10 Oct 2023

Forensic analysis of the Turkey 2023 presidential election reveals extreme vote swings in remote areas

PONE-D-23-19758R1

Dear Dr. Klimek,

We’re pleased to inform you that your manuscript has been judged scientifically suitable for publication and will be formally accepted for publication once it meets all outstanding technical requirements.

Kind regards,

Ali B. Mahmoud, Ph.D.

Academic Editor

PLOS ONE

Additional Editor Comments (optional):

Reviewers' comments:

Reviewer's Responses to Questions

**Comments to the Author**

1. If the authors have adequately addressed your comments raised in a previous round of review and you feel that this manuscript is now acceptable for publication, you may indicate that here to bypass the “Comments to the Author” section, enter your conflict of interest statement in the “Confidential to Editor” section, and submit your "Accept" recommendation.

Reviewer #1: All comments have been addressed

Reviewer #2: All comments have been addressed

Reviewer #3: All comments have been addressed

2. Is the manuscript technically sound, and do the data support the conclusions?

Reviewer #1: Yes

Reviewer #2: (No Response)

Reviewer #3: Yes

3. Has the statistical analysis been performed appropriately and rigorously? 

Reviewer #1: Yes

Reviewer #2: Yes

Reviewer #3: Yes

4. Have the authors made all data underlying the findings in their manuscript fully available?

Reviewer #1: Yes

Reviewer #2: Yes

Reviewer #3: Yes

5. Is the manuscript presented in an intelligible fashion and written in standard English?

Reviewer #1: Yes

Reviewer #2: Yes

Reviewer #3: Yes

6. Review Comments to the Author

Reviewer #1: I find the authors revision of their original submission responsive to all the issues and recommendation I made in my first review. I support publication of the revised manuscript.

Reviewer #2: (No Response)

Reviewer #3: (No Response)

7. PLOS authors have the option to publish the peer review history of their article (what does this mean?). If published, this will include your full peer review and any attached files.

Reviewer #1: **Yes: **Robert M. Stein

Reviewer #2: No

Reviewer #3: No

---

## [Editor Report · Acceptance letter]

20 Oct 2023

PONE-D-23-19758R1 

Forensic analysis of the Turkey 2023 presidential election reveals extreme vote swings in remote areas 

Dear Dr. Klimek:

I'm pleased to inform you that your manuscript has been deemed suitable for publication in PLOS ONE. Congratulations! Your manuscript is now with our production department. 

Kind regards, 

on behalf of

Dr. Ali B. Mahmoud 

Academic Editor

PLOS ONE